# Psychometric Evaluation of the Brief-COPE Inventory and Exploration of Factors Associated with Perceived Stress among Peruvian Nurses

**DOI:** 10.3390/healthcare12171729

**Published:** 2024-08-30

**Authors:** Jhon Alex Zeladita-Huaman, Carmen Cristina Flores-Rodríguez, Roberto Zegarra-Chapoñan, Sugely Julia Carpio-Borja, Eduardo Franco-Chalco, Teresa De Jesús Vivas-Durand, Henry Castillo-Parra, Silas Hildeliza Alvarado-Rivadeneyra, Orfelina Mariñas-Acevedo

**Affiliations:** 1Academic Department of Nursing, Faculty of Medicine, Universidad Nacional Mayor de San Marcos, Lima 15001, Peru; jhonzeladita@hotmail.com (J.A.Z.-H.); tvivasd@unmsm.edu.pe (T.D.J.V.-D.); salvarador@unmsm.edu.pe (S.H.A.-R.); 2School of Nursing, Faculty of Medicine, Universidad Nacional Mayor de San Marcos, Lima 15001, Peru; carmen.flores10@unmsm.edu.pe; 3Faculty of Health Science, Universidad María Auxiliadora, Lima 15408, Peru; 4Nursing Department, Hospital Marino Molina Scippa, Lima 15311, Peru; scarpiob2023@gmail.com; 5Psychology Academic Department, Pontificial Catholic University of Perú, Lima 15008, Peru; a20204683@pucp.edu.pe; 6Faculty of Psychology, Universidad de San Buenaventura, Medellín 050021, Colombia; gerencia@neuromind.net; 7School of Nursing, Universidad Norbert Wiener, Lima 15311, Peru; linita.marinas@gmail.com

**Keywords:** stress, psychometric study, nurses, coping, Brief-COPE, confirmatory factor analysis

## Abstract

Background: This study aimed to analyze the psychometric properties of the Brief-COPE Inventory and to determine its concurrent validity by examining its association with perceived stress among Peruvian nurses. Methods: A psychometric study was conducted with 434 Peruvian nurses to evaluate the psychometric properties of the Brief-COPE Inventory through confirmatory factor analysis. Three stepwise variable selection regression models were implemented. Results: The three-factor model of the Brief-COPE Inventory demonstrated adequate fit indices (root mean square error of approximation = 0.052, standardized root mean square residual = 0.068, and both the comparative fit index and the Tucker–Lewis index = 0.95). Additionally, the factors were significantly correlated (*p* < 0.001), and the reliability was adequate (ω = 0.90). Nurses reported a medium level of perceived stress, with associated factors including having received stress management training, fear of COVID-19, and problem-focused coping strategies (*p* < 0.05). Conclusion: This study confirms that the Brief-COPE Inventory is a valid tool for measuring coping strategies among Peruvian nurses due to its good model fit, excellent reliability, and concurrent validity with perceived stress. However, further research is needed to assess its validity in the specific areas of performance perceived by nursing professionals.

## 1. Introduction

Health professionals increasingly face more stressful situations as a result of internal factors and persistent challenges in the modern healthcare system [1,2]. Specifically, nurses faced high levels of stress and burnout during and after the COVID-19 pandemic due to work overload, staffing shortages, and health issues [1]. The few studies that have analyzed COVID-19-related perceived stress in health students and professionals highlight its association with sociodemographic variables such as gender, age, religion [3,4], years of work experience, work area [5], personal and family history of epidemiological infection [6], fears, worries, and coping strategies [7].

Coping is defined as a psychological construct that groups together a wide variety of thoughts, feelings, and behaviors that individuals use to manage internal or external demands from stressful situations that exceed their own skills [8,9,10]. Coping strategies comprise specific actions aimed at modifying the stimulus that a stressful situation generates and controlling the emotions that this stimulus produces in individuals [8,9,11].

Initially, Folkman and Lazarus [12] proposed the classification of coping strategies based on the coping focus: problem-focused coping (strategies aimed at actively solving stressful situations) and emotion-focused coping (strategies aimed at managing or reducing the emotions and feelings caused by stressful situations). Subsequently, Carver et al. [13] proposed a distinction based on how they are used: approach coping (strategies aimed at actively coping with stress or related emotions) and avoidance coping (strategies aimed at avoiding stressful situations). Finally, taking into account the outcomes of using coping strategies, we have adaptive coping (more likely leading to desired and favorable outcomes for health) and maladaptive coping (leading to possible benefits in the short term but more likely to undesired outcomes in the long term) [14]. However, the outcome of a coping strategy in this last classification varies among individuals and according to their context [13,15]. Thus, nurses can cope with a stressful situation and its emotional consequences by actively and directly approaching or avoiding problems [16].

Based on the aforementioned theoretical models, several scales have been developed to measure coping strategies. The two most commonly used scales are the Ways of Coping Questionnaire (WCQ) developed by Folkman and Lazarus [17] and the Coping Orientation to Problems Experienced (COPE) designed by Carver et al. [13]. Likewise, scales have been developed to measure situational or stress-related coping strategies; these include the Pain Coping Questionnaire validated by Reid et al. [18] to assess the pain coping strategies of children and adolescents; the Multidimensional Coping Inventory proposed by Endler and Parker [19]; and the Coping Inventory for Stressful Situations suggested by Endler and Parker [20]. Although valid and reliable in the sample studied, they are very long scales (39 to 66 items), which would not favor the measurement of associated variables and could limit their use in long research protocols.

One of the most widely used scales to measure coping strategies is the Brief-COPE Inventory (BCI). This multidimensional scale made up of 52 items (14 scales) was originally developed by Carver et al. [13]. Subsequently, based on the theoretical model of Folkman and Lazarus’s WCQ and the model of behavioral self-regulation, the BCI was reduced to 28 items (14 scales). In the healthcare context, numerous psychometric studies have empirically proved that the BCI has adequate psychometric properties for measuring coping strategies in patients in India [21] and China [22], family caregivers in the United Kingdom [23], and health professionals in Italy [15]. It is worth noting that the range of items and the factor structure vary widely.

Similarly, during the COVID-19 pandemic, several psychometric studies were conducted to validate the BCI with nurses from different countries. In the United Arab Emirates, Abdul Rahman et al. [1] found that this scale presented inadequate fit indexes when evaluating four previous models; therefore, they validated a version of 22 items grouped into two factors, using a second-order model. Likewise, in Italy, Bongelli et al. [15] carried out psychometric studies that reported the validity of a scale of 21 items (7 were eliminated) grouped into six factors (McDonald’s omega = 0.811). Scales validated with nurses reported varied compositions.

Although the BCI is one of the most used scales thanks to its adequate psychometric properties in different contexts and populations, it has not been validated with nurses in Latin America. The purpose of this study is to analyze the factorial structure of the BCI in a sample of Peruvian nurses, as no psychometric studies were found in a context like the Peruvian one with health workers. Additionally, this study aims to update this scale to reflect social and cultural changes in the context of the pandemic, to evaluate the cross validation of the BCI with scales that measure stress, and to explore new factors associated with stress caused by the COVID-19 pandemic that were not previously considered. Therefore, this study aims to analize the psychometric properties of the BCI—which assesses coping strategies—and to determine its concurrent validity by examining its association with perceived stress among Peruvian nurses using models that incorporate sociodemographic variables and fear of COVID-19.

## 2. Materials and Methods

### 2.1. Sample and Procedures

We conducted a cross-sectional psychometric study with 434 Peruvian nurses to determine the psychometric properties of the BCI by means of a confirmatory factor analysis (CFA). In addition, we implemented three stepwise variable selection regression models to analyze the factors associated with perceived stress and its dimensions.

The study sample consisted of nurses working in health facilities across different cities in Peru, all of whom had access to an electronic device with internet. It excluded nurses who only held a teaching position. We set the minimum sample size according to the classic criterion of 10 participants for each item of the questionnaire [24], for a total of 300 participants. A non-probabilistic convenience sampling method was used to select the sample until it reached 450 responses, aiming to increase statistical power and reduce the impact of individual biases. This approach was chosen because the main purpose of this study was to evaluate the psychometric properties of the BCI. Additionally, to achieve greater diversification, participants from both public and private health facilities across different regions of Peru were invited.

Data collection was conducted through a virtual Google Forms questionnaire between November 2021 and February 2022, during the third wave of COVID-19 infections in Peru. The strategies employed were as follows: distributing the virtual questionnaire through social media (WhatsApp groups), posting invitations to participate in the study on Facebook pages, and sending emails to nurses from various hospitals and those pursuing postgraduate studies.

After participants provided informed consent, they were asked about their sociodemographic information (age, gender, marital status) and job-related information (work area, years of work experience, whether they had received stress management training, and whether they had taken care of COVID-19 patients during the last month).

### 2.2. Measurement Tools

#### 2.2.1. Brief-COPE Inventory

The BCI is based on the COPE [13], and it was validated by expert judgment with the participation of Peruvian nurses involved in health care and university teaching activities. After calculating the concordance index according to the content validity coefficient by [25], we found good concordance (CVC = 0.8250). Subsequently, we conducted a pilot test with 65 nurses and obtained an adequate reliability (Cronbach’s alpha = 0.910) [26]. The 30-item Likert scale has four response alternatives ranging from 1 (seldom) to 4 (almost always).

#### 2.2.2. Fear of COVID-19 Scale

We used the Fear of COVID-19 Scale, validated in the Peruvian population through a psychometric study, which reports a bifactorial model with adequate goodness-of-fit indices (comparative fit index (CFI) = 0.988, root mean square error of approximation (RMSEA) = 0.075) and invariance according to the type of healthcare worker and age (ΔCFI < 0.01) [27]. Its response alternatives range between 1 (strongly disagree) and 5 (strongly agree). Thus, the higher the score on the scale, the higher the fear. Another study carried out with Peruvian nurses reported that this scale has adequate reliability (McDonald’s omega coefficient = 0.87) [28].

#### 2.2.3. Perceived Stress Scale

The COVID-19-Related Perceived Stress Scale (PSS-10-C) was utilized, which was developed in 2020 among Colombian nurses for use in the context of the COVID-19 pandemic. Through an exploratory factor analysis, it was reported to consist of a single factor with a Cronbach’s alpha = 0.86. Additionally, Bartlett’s test showed χ^2^ = 1399.35; degrees of freedom = 54; *p* < 0.001; and Kaiser–Meyer–Olkin = 0.82 [29]. Another psychometric study conducted on a Spanish population confirmed its unidimensional internal structure (χ^2^ = 62.67; df = 35; *p* = 0.003; RMSEA = 0.05; standardized root mean square residual (SRMR) = 0.04; CFI = 0.99; Tucker–Lewis index (TLI) = 0.99), its construct validity, and invariance by gender [30]. Subsequently, the author who initially designed the scale, through another psychometric study conducted on 1,136 university students in Colombia, reported that it has adequate reliability (Cronbach’s alpha = 0.85) and is composed of two factors (χ^2^ = 295.6; df = 34; *p* < 0.001; χ^2^/df = 8.7; RMSEA = 0.08; 90% CI, 0.07–0.09; CFI = 0.93; TLI = 0.91; SRMR = 0.05) [31]. The first factor, “distress” (Cronbach’s alpha = 0.83), consists of 5 items that assess how a person perceives and evaluates stressors, considering the threat or difficulty they represent to their well-being (e.g., Item 1: “I have felt affected as if something serious were going to happen unexpectedly due to the epidemic”). The second factor, “coping” (Cronbach’s alpha = 0.77), consists of five items that assess how a person manages, reduces, or tolerates the demands they consider stressful (e.g., Item 4: “I have been confident in my ability to handle my problems related to the epidemic”) [31]. This Likert-type scale includes five response options, ranging from 1 (never) to 5 (always), with four of the items requiring reverse coding. In this study, we determined stress levels (high, medium, and low) using the 33rd and 66th percentiles.

### 2.3. Analysis

To meet the objectives of this study, we first conducted a reliability analysis of the BCI using confirmatory factor analysis (CFA). We opted not to perform an exploratory factor analysis (EFA) because CFA is suitable when a theoretical structure based on the conceptual framework is already established [32,33]. Given the ordinal nature of the data, we employed weighted least squares means and variances in the CFA. Considering the use of non-continuous indicators, the fit criteria only considered an SRMR lower than 0.07 [34]. The initial model did not fit satisfactorily (SRMR = 0.074). Consequently, we allowed residuals for items 19, 20, and 23 to correlate, as these items were associated with the participants’ religious criteria, suggesting the presence of residual variance not necessarily related to coping strategies. The final model, along with its fit indicators, is presented in the Results section. We calculated the McDonald’s omega coefficient (ώ) based on the estimated confirmatory model to assess the instrument’s reliability.

Furthermore, to determine the variables associated with perceived stress in nurses, we conducted three series of stepwise variable selection regression models as follows: the first series for the total perceived stress scale; the second series for Factor 1, “distress”; and the third series for Factor 2, “coping”. To be more specific, we performed these three types of analyses. In addition to examining the relationship between various coping strategies and the total perceived stress scale, we analyzed the two factors separately to observe their association with different coping strategies.

In the three cases, we entered the variables to the model following a three-step sequence: in the first step, the sociodemographic variables (age, marital status, work area, years of work experience, taking care of COVID-19 patients, and having received stress management training); in the second step, the fear of COVID-19 predictor; and in the third step, the dimensions of coping strategies (problem-focused coping, emotion-focused coping, and avoidance coping). Subsequently, we evaluated the significance of the increase in explained variance (R2) with the purpose of comparing the relative fit of these models. Regarding the assumptions of the models, we assessed the normality and homogeneity of variances and the multicollinearity for all cases, and we were able to confirm that these assumptions were correct. Finally, we checked Cook’s D indicators for possible outliers and we identified none in any of the models. We ran all these statistical analyses in R v4.2.1 software.

## 3. Results

After verifying the filled-out questionnaires, we eliminated sixteen of them because they had at least one unanswered question. Accordingly, the sample for analysis comprised 434 nurses, predominantly women (90.55%). The mean age of the participants was 39.68 years old (SD = 10.17). Table 1 shows the other characteristics of the sample.

The analysis of the correlation matrix in Table 2 reveals several significant relationships between stress perception, coping strategies, and fear of COVID-19. Notably, stress perception is strongly positively correlated with fear of COVID-19, indicating that individuals who perceive higher levels of stress are likely to experience increased fear of the pandemic. Conversely, the second factor of stress, “coping”, is significantly negatively correlated with fear of COVID-19. Among coping strategies, these dimensions are moderately to strongly correlated with each other. Fear of COVID-19 is strongly positively correlated with total perceived stress, reinforcing the link between high stress levels and pandemic-related fear. Additionally, there is a weak but significant positive correlation between avoidance coping and fear of COVID-19, suggesting that avoidance behaviors may contribute to heightened fear during the pandemic.

### 3.1. Confirmatory Factor Analysis of the Coping Strategies Questionnaire

Figure 1 shows the factor model of the BCI. According to the fit indicators of this model, an adequate fit was achieved by allowing residual variances of items 19, 20, and 23 to correlate with each other. Particularly, we observed that the SRMS was 0.068. When observing the factor loadings of the items on each of the estimated dimensions, we can see that all of them showed standardized factor loadings greater than 0.26, indicating that all the items had considerable loadings on the estimated factors. The correlated residual variances were all significant and higher than 0.37, indicating that items 19, 20, and 23 share a variance that cannot be explained by emotion-focused coping. As for the correlation of coping dimensions, we can observe that problem-focused coping had a very strong and positive correlation with emotion-focused coping (r = 0.74) while keeping a strong correlation with avoidance coping (r = 0.39). For its part, emotion-focused coping had a very strong and positive correlation with avoidance coping (r = 0.53). Lastly, the McDonald’s omega coefficient resulting from the factors was 0.87 for problem-focused coping, 0.91 for emotion-focused coping, and 0.90 for avoidance coping, which indicated that the instrument had an overall excellent internal consistency.

Regarding perceived stress, 36.41% of the sample (158 participants) presented a medium level; 34.79% (151 participants), a low level; and only 28.0% (125 participants), a high level.

### 3.2. Predictive Models of Perceived Stress

Table 3 shows the series of stepwise regression models for the prediction of total perceived stress. More specifically, we can point out that the first model—which only included sociodemographic variables—explained 5% of variance (R2 = 0.05, *p* < 0.001). For its part, the model that included the fear of COVID-19 predictor showed an increase in explained variance by 21% (Δ R2 = 0.21, *p* < 0.001). Finally, the model that included coping strategies presented a 1% increase in explained variance (Δ R2 = 0.01, *p* = 0.037), and for this reason, we interpreted this third model as indicated below.

In this model, we identified that older patients present a higher stress perception score (B = −0.15, *p* = 0.026). Similarly, we observed that men showed lower levels of perceived stress than women did (B = −0.10, *p* = 0.018). In addition, people who had received stress management training had lower levels of perceived stress than those who had not (B = −0.12, *p* = 0.007). We also noted that participants who showed higher levels of fear of COVID-19 also showed higher levels of perceived stress (B = 0.45, *p* < 0.001). Lastly, participants who reported higher levels of problem-focused coping also reported lower perceived stress scores (B = −0.15, *p* = 0.002). 

Table 4 shows the stepwise regression models estimated for predicting the first stress factor, “distress”. Specifically, we can see that the first model including only the sociodemographic variables explained 3% of variance (R2 = 0.03, *p* = 0.005). Then, the model in which we entered the fear of COVID-19 predictor presented a 25% increase in explained variance (Δ R2 = 0.28, *p* < 0.001). Finally, the model including coping strategies did not reveal a significant increase in explained variance (Δ R2 = 0.00, *p* = 0.341). Therefore, we interpreted the model of the second step, without coping strategies.

In the model of the second step, we identified that older patients present a higher perception score of the first stress factor “distress” (B = −0.14, *p* = 0.032). Similarly, men reported lower levels of stress perception than women did (B = −0.12, *p* = 0.004). Finally, we observed that participants who reported higher levels of fear of COVID-19 also showed higher levels of “distress” (B = 0.50, *p* < 0.001). No other variables were shown to be significantly associated with perceived stress.

In the second-step model, we observed a significant negative association between between age and the perception score of the first stress factor “distress” (B = −0.14, *p* = 0.032). Additionally, men reported lower levels of stress perception compared to women (B = −0.12, *p* = 0.004). We also observed that participants who reported higher levels of fear of COVID-19 exhibited higher levels of “distress” (B = 0.50, *p* < 0.001). There were no other variables that were found to have a significant association with perceived stress.

Table 5 shows the stepwise regression models estimated to predict the second stress factor, “coping.” It can be observed that the first model explained 3% of variance (R2 = 0.03, *p* = 0.012). The second model, in which we included fear of COVID-19 as a predictor, increased explained variance by only 4% (Δ R2 = 0.04, *p* < 0.001). The third model, in which we entered coping strategies, also increased explained variance by 4% (Δ R2 = 0.04, *p* < 0.001). Considering this result, we interpreted this third model.

Particularly, this model showed that nurses working in the emergency department reported higher levels of stress coping than those working in the intensive care unit (B = 0.11, *p* = 0.047). Similarly, those who had received stress management training had higher levels in the stress coping dimension than their counterparts who had not received such training (B = −0.11, *p* = 0.015). Likewise, participants who reported higher levels of fear of COVID-19 showed lower levels of the factor “coping” (B = −0.21, *p* < 0.001). Lastly, participants who reported higher levels of problem-focused coping also showed higher scores on the factor “coping” (B = 0.21, *p* < 0.001). We observed no other significant effects in the present model. Similarly, participants who reported higher levels of fear of COVID-19 exhibited lower levels of the “coping” factor (B = −0.21, *p* < 0.001). Conversely, participants who reported higher levels of problem-focused coping also demonstrated higher scores on the “coping” factor (B = 0.21, *p* < 0.001). No other significant effects were observed in this model.

## 4. Discussion

This study provides important evidence of the adequate psychometric properties of the BCI in a sample of Peruvian nurses. It demonstrates concurrent validity because of the association found with perceived stress, thus providing substantial support for its use in assessing coping strategies in this population. This finding is consistent with previous studies that validated BCI with different populations and contexts, including healthcare professionals in Italy [15] and family caregivers in the United Kingdom [23], as well as specific stressful contexts such as the COVID-19 pandemic [1].

The CFA supported the presence of three factors or dimensions of the inventory: problem-focused coping, emotion-focused coping, and avoidance coping. This finding is in line with the theory underlying the coping model of Folkman and Lazarus [12], as well as with the distinctions proposed by Carver et al. [13] regarding how coping strategies are used. However, it differs from other studies where different dimensions of coping strategies were found [1,15]. These differences between studies conducted in different contexts suggest that coping strategies are strongly dependent on cultural and contextual factors, which makes studies such as the present one necessary to have context-specific validated instruments.

In the same vein, it is important to note that the confirmatory model showed a good fit only when the residual variances of three items related to religion as a coping strategy were correlated. This finding suggests that religion may play a crucial role in stress coping among Peruvian nurses, which is consistent with previous studies that have highlighted the importance of religion and spirituality in stress coping among various populations [5,35,36]. Overall, these studies indicate that religion can provide emotional support, meaning, and a sense of resilience that help nurses cope with perceived stress and thrive in their profession. However, it is important to note that these items had significant and strong factor loadings on the emotion-focused coping factor, demonstrating a correlation between spirituality and emotions as a coping mechanism.

Similarly, the association between stress and coping strategies reported in this study is consistent with several studies conducted among nurses in Korea [37] and China [38], nursing students in Colombia [3], and people in general in China [39]. Moreover, the regression analysis showed that participants with higher levels of problem-focused coping had lower stress scores, which is in line with a study conducted with nurses working in psychiatric hospitals in Egypt [40]. This suggests that when nursing professionals cope with and actively address stressors, their stress levels could be significantly reduced [38].

The other predictors of perceived stress found in the regression analysis were gender, having received stress management training (which only explained 5% of the variance), and fear of COVID-19. These findings are consistent with numerous studies reporting associations between stress and gender [7] and stress and fear [41,42,43]. However, it contrasts with a study that reports that gender is not a predictor of stress in nurses [44]. These associations highlight the need to target educational interventions towards nurses who report high levels of fear. Programs aimed at strengthening the ability to cope with stress through the use of various self-management techniques and cognitive strategies would help enhance nurse participation and establish protocols for stress prevention. Additionally, it would enable nurses to adopt a proactive and positive approach to directly address stressors.

In this study, the analysis of the association between coping strategies and the overall perception of stress and each specific factor is based on the following key aspects: 1. Stress is complex and has many facets, including emotional, cognitive, behavioral, and physiological. These measures can affect people’s health and well-being in multiple ways. Therefore, these three types of analysis allow for a more complete and refined understanding of how coping mechanisms relate to the experience of stress. 2. The role of different types of coping strategies varies. In the case of this study, we found that only problem-focused coping strategies were associated with both the total stress scale and the coping factor but not with the distress factor. Thus, analyzing each factor allowed for the identification of the most relevant coping techniques for managing each specific aspect of stress. 3. The design of more specific and effective psychological interventions may depend on understanding how coping relates to the different dimensions of stress.

Assessing health professionals’ perceived stress and coping skills is of great importance from a neuroscience and mental health perspective. Numerous scientific studies have demonstrated that the work environment of health professionals, such as physicians and nurses, is associated with high levels of stress attributed to the demands and pressure of patient care and critical decision making during crisis situations [37,38]. Consequently, a thorough and current assessment of these factors can offer valuable insights into the emotional well-being and coping skills of such professionals, which may play a crucial role in the quality of patient care they provide and their own mental health and well-being.

Perceived stress related to the COVID-19 pandemic has been a significant factor in health professionals’ lives, particularly frontline nurses. Assessing coping strategies and their associated factors among these professionals is critical to understanding how they deal with the challenges of this unusual context. Therefore, designing and validating a psychometrically sound scale could have substantial implications for psychology and nursing practice.

This study highlights the importance of validating psychometric instruments across different cultures. While Folkman and Lazarus’s stress coping model [17] has been accepted in various contexts, confirming its dimensions in a sample of Peruvian nurses demonstrates how cultural variables influence the use of coping strategies. This theoretical finding implies that universal coping models may require modifications or adaptations for different cultural and population contexts. Furthermore, the validation of the three dimensions of the BCI (problem-focused coping, emotion-focused coping, and avoidance) not only confirms but also expands the existing knowledge on the classification of coping strategies. This suggests that coping theories should consider the adaptability of strategies to cultural and professional contexts, potentially leading to theoretical revisions in future research.

From a practical perspective, identifying predictors of stress such as gender, professional experience, and fear of COVID-19 suggests that stress management interventions for healthcare professionals should be customized to address individual needs. Stress management training programs should consider these specific factors to be more effective in reducing stress among nurses, thereby improving their well-being and the quality of care they provide. Furthermore, this study emphasizes the importance of healthcare professionals continuously evaluating their coping strategies. Implementing validated psychometric tools, such as the BCI, in hospital settings can help identify professionals who need support at an early stage, allowing for more timely and effective interventions. These implications underscore the need to adapt both theory and practice to the specific circumstances of healthcare professionals to enhance their well-being and productivity at work.

This study has some limitations. First, given that data were collected through self-reporting, the results could be affected by common method and social desirability bias, thus requiring that the reported findings be verified by additional studies. The second limitation of this study is that by using non-probability sampling, the study results cannot be generalized to the entire Peruvian population. Finally, since we only performed the CFA based on a theoretical framework, other studies are required to confirm this structure in studies that consider different samples.

## 5. Conclusions

This psychometric study demonstrated that a three-factor BCI is a valid tool to measure coping strategies in Peruvian nurses. It showed an adequate fit to the model (RMSEA = 0.052, SRMS = 0.068, CFI = 0.95, and TLI = 0.95), excellent reliability (ώ = 0.90), and concurrent validity with perceived stress.

Nursing professionals reported medium levels of perceived stress. Regarding the predictive model, the first analysis (5% of explained variance) found gender and stress management training to be predictor variables. Likewise, in the second analysis (26% of explained variance), fear of COVID-19 and professional experience were also found to be predictors. Lastly, in the third analysis (27% of explained variance), the problem-focused coping strategies score was reported as a predictor factor.

## Figures and Tables

**Figure 1 healthcare-12-01729-f001:**
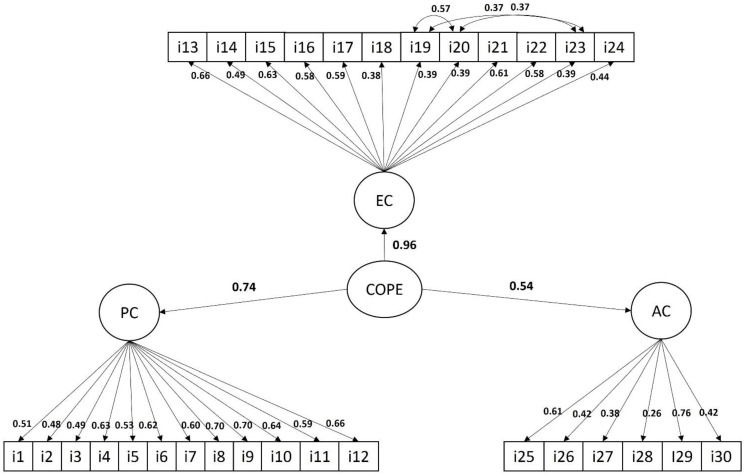
Factor model of the coping strategies questionnaire. Note: PC: problem-focused coping; EC: emotion-focused coping; AC: avoidance coping. All model parameters are significant at *p* < 0.001.

**Table 1 healthcare-12-01729-t001:** Sociodemographic and job-related characteristics of the sample.

	f	%
Gender		
Women	393	90.55
Men	41	9.45
Marital status		
Single	187	43.09
Married	205	47.24
Divorced	42	9.68
Work area		
Intensive care unit	116	26.73
Emergency care	62	14.29
Hospitalization	110	25.35
Primary health care	34	7.83
COVID-19 triage	36	8.29
Other areas *	76	17.51
Years of work experience		
Less than 5	104	23.96
5 to 10	121	27.88
More than 10	209	48.16
Received stress management training		
No	206	56.90
Yes	228	43.10
Took care of COVID-19 patient		
No	119	27.42
Yes	315	72.58

* Neonatology, surgery, among others.

**Table 2 healthcare-12-01729-t002:** Correlation matrix between dimensions of perceived stress, dimensions of coping strategies, and fear of COVID-19.

	1	2	3	4	5	6
1. First stress factor: “distress”						
2. Second stress factor: “coping”	−0.3 **					
3. Perceived stress (total)	0.85 **	−0.75 **				
4. Problem-solving coping	−0.03	0.24 **	−0.15 *			
5. Emotion-focused coping	0.12 *	0.13 *	0.01	0.59 **		
6. Avoidance coping	0.15 *	0.02	0.09 *	0.24 **	0.37 **	
7. Fear of COVID-19	0.51 **	−0.23 **	0.48 **	−0.06	0.07	0.17 **

* *p* < 0.05, ** *p* < 0.001.

**Table 3 healthcare-12-01729-t003:** Regression models for predicting total perceived stress.

	Step 1		Step 2		Step 3	
	*B*	*p*	*B*	*p*	*B*	*p*
Age	−0.13	0.080	−0.15	0.020	−0.15	0.026
Gender (men)	−0.14	0.004	−0.12	0.005	−0.10	0.018
Marital status (married)	0.05	0.391	0.01	0.881	0.01	0.860
Marital status (divorced)	0.02	0.694	−0.02	0.759	−0.02	0.711
Area (emergency care)	−0.09	0.101	0.06	0.258	−0.06	0.228
Area (hospitalization)	−0.05	0.401	−0.02	0.640	−0.02	0.654
Area (primary care)	−0.04	0.397	−0.01	0.879	−0.01	0.980
Area (triage)	−0.04	0.420	−0.01	0.789	−0.01	0.770
Area (other areas)	−0.03	0.578	0.03	0.560	0.03	0.535
Experience (5 to 10 years)	0.07	0.300	0.10	0.092	0.10	0.088
Experience (>10 years)	0.15	0.114	0.02	0.043	0.16	0.053
Took care of COVID-19 patients (Yes)	0.08	0.087	0.07	0.133	0.06	0.152
Stress management training (yes)	−0.19	<0.001	−0.12	0.005	−0.12	0.007
Fear of COVID-19	-	-	0.46	<0.001	0.45	<0.001
Problem-focused coping	-	-	-	-	−0.15	0.002
Emotion-focused coping	-	-	-	-	0.06	0.288
Avoidance coping	-	-	-	-	0.01	0.930
R2	0.05 *		0.26 *		0.27 *	
Delta R2	-		0.21*		0.01 **	

Note: *B* = standardized coefficient. The reference category for gender is women; for marital status, single; for work area, intensive unit care; and for experience, less than 5 years. * *p* < 0.001, ** *p* < 0.01.

**Table 4 healthcare-12-01729-t004:** Regression models for predicting the first stress factor, “distress”.

	Step 1		Step 2		Step 3	
	*B*	*p*	*B*	*p*	*B*	*p*
Age	−0.11	0.130	−0.14	0.032	−0.13	0.040
Gender (men)	−0.15	0.003	−0.12	0.004	−0.11	0.010
Marital status (married)	0.09	0.080	0.05	0.264	0.05	0.329
Marital status (divorced)	0.01	0.861	−0.03	0.531	−0.03	0.587
Area (emergency care)	−0.04	0.526	0.01	0.982	−0.01	0.987
Area (hospitalization)	0.03	0.580	0.06	0.244	0.05	0.339
Area (primary care)	−0.01	0.878	0.03	0.479	0.03	0.520
Area (triage)	−0.01	0.823	0.02	0.648	0.02	0.720
Area (other areas)	−0.01	0.896	0.06	0.237	0.04	0.386
Experience (5 to 10 years)	0.01	0.832	0.05	0.413	0.05	0.416
Experience (>10 years)	0.12	0.193	0.14	0.776	0.14	0.078
Took care of COVID-19 patients (yes)	0.09	0.070	0.07	0.105	0.62	0.144
Stress management training (yes)	−0.15	0.002	−0.07	0.799	−0.77	0.068
Fear of COVID-19	-	-	0.50	<0.001	0.49	<0.001
Problem-focused coping	-	-	-	-	−0.05	0.325
Emotion-focused coping	-	-	-	-	0.09	0.119
Avoidance coping	-	-	-	-	0.02	0.600
R2	0.03 *		0.28 **		0.28 **	
Delta R2	-		0.25 **		0.00	

Note: *B* = standardized coefficient. The reference category for gender is women; for marital status, single; for work area, intensive unit care; and for experience, less than 5 years. * *p* < 0.01, ** *p* < 0.001.

**Table 5 healthcare-12-01729-t005:** Regression models for predicting the second stress factor, “coping”.

	Step 1		Step 2		Step 3	
	*B*	*p*	*B*	*p*	*B*	*p*
Age	0.10	0.206	0.11	0.151	0.10	0.164
Gender (men)	0.07	0.139	0.06	0.188	0.05	0.316
Marital status (married)	0.04	0.506	0.05	0.314	0.04	0.410
Marital status (divorced)	−0.03	0.623	−0.01	0.844	−0.01	0.996
Area (emergency care)	0.12	0.030	0.10	0.054	0.11	0.047
Area (hospitalization)	0.13	0.026	0.12	0.038	0.10	0.066
Area (primary care)	0.07	0.180	0.05	0.297	0.04	0.437
Area (triage)	0.06	0.235	0.05	0.341	0.05	0.369
Area (other areas)	0.05	0.399	0.02	0.713	−0.01	0.965
Experience (5 to 10 years)	−0.10	0.107	−0.11	0.064	−0.12	0.056
Experience (>10 years)	−0.11	0.221	−0.12	0.178	−0.11	0.225
Took care of COVID-19 patients (yes)	−0.04	0.418	−0.03	0.513	−0.03	0.475
Stress management training (yes)	0.16	<0.001	0.13	0.007	0.11	0.015
Fear of COVID-19	-	-	−0.21	<0.001	−0.21	<0.001
Problem-focused coping	-	-	-	-	0.21	<0.001
Emotion-focused coping	-	-	-	-	0.00	0.993
Avoidance coping	-	-	-	-	0.02	0.656
R2	0.03 *		0.07 **		0.11 **	
Delta R2	-		0.04 **		0.04 **	

Note: *B* = standardized coefficient. The reference category for gender is women; for marital status, single; for work area, intensive unit care; and for experience, less than 5 years. * *p* < 0.05, ** *p* < 0.001.

## Data Availability

The original data presented in the study are openly available in: https://doi.org/10.6084/m9.figshare.24319207; Questionnaires used in this study, Spanish version: https://doi.org/10.6084/m9.figshare.24319219.v1; Questionnaires used in this study, English version: https://doi.org/10.6084/m9.figshare.24319210.v1.

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
