# Peer review of "Psychometric Evaluation of the Brief-COPE Inventory and Exploration of Factors Associated with Perceived Stress among Peruvian Nurses"

_healthcare, 2024, doi:10.3390/healthcare12171729_

Round 1

Reviewer 1 Report

Comments and Suggestions for Authors

The present manuscript describes the psychometric of the Brief Coping Inventory among Peruvians nurses. Authors provide factor analyses and associations with external variables to examine the nomological net. Overall, the study is sound, but there are some points that should be addressed in a revision of the manuscript. 

(1) In line with APA guidelines, please change "sex" to "gender" and "females" to "women" and "males" to "men."

(2) In the results section, please report the fit indexes for the initial model (i.e., the model without correlations between items). Since no cut-off recommendations are available for CFA models using the WLSMV estimator, it is possible that the fit for the basic model is already sufficient. 

(3) Subsequent regression analyses assume a total score of the scale, but the assumption of a total score in terms of a general factor is not tested in the CFA. Please compute a second-order and/or bifactor model to clarify whether a total score can be assumed. 

(4) Please provide a table containing the correlations between the study variables. 

Author Response

Dear Reviewer, we sincerely appreciate your comments and suggestions, which have greatly helped us improve our manuscript, especially in methods and results sections. Below, we present our responses to each of your observations:

Comment 1: The present manuscript describes the psychometric of the Brief Coping Inventory among Peruvians nurses. Authors provide factor analyses and associations with external variables to examine the nomological net. Overall, the study is sound, but there are some points that should be addressed in a revision of the manuscript.

In line with APA guidelines, please change "sex" to "gender" and "females" to "women" and "males" to "men."

Response 1: Thank you for your insightful observation. We agree with your suggestion and have changed these words according to the recommendations.

Comment 2: In the results section, please report the fit indexes for the initial model (i.e., the model without correlations between items). Since no cut-off recommendations are available for CFA models using the WLSMV estimator, it is possible that the fit for the basic model is already sufficient.

Response 2: Agree, that although there are no specific recommendations, we prefer to maintain the rigor of the recommendations given for Maximum Likelihood estimators. Additionally, Model fit indicators without correlated variations were added. All these updates can be found in 2.3. Analysis lines 179 to 180.

Comment 3: Subsequent regression analyses assume a total score of the scale, but the assumption of a total score in terms of a general factor is not tested in the CFA. Please compute a second-order and/or bifactor model to clarify whether a total score can be assumed.

Response 3: This is an interesting comment. However, in the regression analysis, we did not use the total score of the Brief-COPE Inventory. In Table 2, the Perceived Stress Scale was used, which we are not validating in this study, nor did we calculate a factor model. The instrument has been reported to have adequate psychometric properties, so this model is omitted.

Comment 4: Please provide a table containing the correlations between the study variables.

Response 4: We agree with your suggestion and have added a table of Correlations with Stress, Fear, and Coping Strategies. This additional information can be found in 3. Result section, table 2, lines 213 to 225.

We once again thank you for your valuable suggestions and are confident that these improvements will significantly strengthen our work. We remain attentive to any additional comments you may have.

Sincerely,

Roberto Zegarra.

Reviewer 2 Report

Comments and Suggestions for Authors

it is a valuable topic to explore psychometric properties of the Brief-COPE Inventory and factors associated with COVID-19 related perceived stress in Peruvian nurses. However, some areas in the manuscript could still be improved. 1.The study simply measured the coping patterns of Peruvian nurses using the Brief-COPE inventory and explored the impact of coping on their perceived stress. This does not seem to fit with the title (which suggests a scale revision of the Brief-COPE inventory and exploration on a range of factors associated with perceived stress). 2. The Brief-COPE inventory (BCI) has been extensively validated, how does this study differ from previous studies? The value of this study needs to be stated more clearly. In the introduction section it is suggested that authors list possible contributions. 3.What is the relationship between the two parts: “psychometric properties of the Brief-COPE Inventory” and “factors associated with COVID-19-related perceived stress”? This study did not examine the widely recognized factors that influence stress. 4. How were respondents selected? What sampling method was used? It is recommended that the authors add a clear description. 5. The presentation of the Perceived Stress Scale is confusing. “Perceived Stress Scales scale comprises two dimensions: stress perception (five items) and stress coping (five items).” What is the difference between perceived stress and stress perception? 6. It is recommended that at least one item be given after each scale. 7. How did the pressure scale reflect COVID-19-related perceived stress? 8. Reliability and validity tests need to be supplemented. 9. In the results section, what is the purpose of analyzing the impact of coping on overall perceptions of stress and each dimension separately? This is not a usual way of reporting. 10. The discussion section suggests an exploration of both theoretical and practical implications.

Author Response

Dear Reviewer, we would like to express our sincere gratitude for the time you have dedicated to reviewing our manuscript and for the valuable insights, recommendations, and suggestions you have provided. These contributions have significantly improved the quality and clarity of our scientific manuscript. Below, we present our responses to each of your observations:

Comment 1: It is a valuable topic to explore psychometric properties of the Brief-COPE Inventory and factors associated with COVID-19 related perceived stress in Peruvian nurses. However, some areas in the manuscript could still be improved.

The study simply measured the coping patterns of Peruvian nurses using the Brief-COPE inventory and explored the impact of coping on their perceived stress. This does not seem to fit with the title (which suggests a scale revision of the Brief-COPE inventory and exploration on a range of factors associated with perceived stress).

Response 1: In response to your suggestion, we have modified the title to Psychometric Evaluation of the Brief-COPE Inventory and Exploration of a Range of Factors Associated with Perceived Stress in Peruvian Nurses. However, we would like to highlight that in this study, we have verified the validity of the instrument in a context where this type of study had not been previously conducted. We also evaluated the criterion validity of the study by examining how it relates to perceived stress, a variable previously associated with coping styles. The objective was also modified.

Comment 2: The Brief-COPE inventory (BCI) has been extensively validated, how does this study differ from previous studies? The value of this study needs to be stated more clearly. In the introduction section it is suggested that authors list possible contributions.

Response 2: We agree with this comment. This is a strength of the study, we are taking an instrument, widely used and known worldwide, that had not been validated in a context like the Peruvian one with health workers. Particularly, this is the gap in the literature that is intended to be addressed. In addition, the fact that the sample is Peruvian is added to the title to give more solidity. The possible contributions of the study are detailed in the introduction section, lines 94 to 99.

Comment 3: What is the relationship between the two parts: “psychometric properties of the Brief-COPE Inventory” and “factors associated with COVID-19-related perceived stress”? This study did not examine the widely recognized factors that influence stress.

Response 3: This comment was also addressed in the response to the first comment earlier in this letter. It should be highlighted that we analyzed the relationship between coping strategies and stress to determine criterion validity, and to have greater statistical control, we added other factors related to stress to the statistical model.

Comment 4: How were respondents selected? What sampling method was used? It is recommended that the authors add a clear description.

Response 4: We have addressed this comment by clarifying the wording regarding the type of sampling used and the strategies for data collection. These updates can be found in 2.1. Sample and Procedures section, lines 115 to 120.

Comment 5, 6 y 7: The presentation of the Perceived Stress Scale is confusing. “Perceived Stress Scales scale comprises two dimensions: stress perception (five items) and stress coping (five items).” What is the difference between perceived stress and stress perception?

It is recommended that at least one item be given after each scale.

How did the pressure scale reflect COVID-19-related perceived stress?

Response 5, 6 y 7: We reiterate our gratitude for these pertinent comments. We have clarified the wording and provided additional information on the validity and reliability of the scale reported in previous studies, the description of what the items of the two factors assess, and described at least one item for each factor. This action allowed us to explain the scale that developed during the COVID-19 pandemic. Also, to avoid confusion, the names of each factor were changed according to the study that validated this scale. These updates can be found in 2.2. Measurement Tools section, lines 150 to 171.

Comment 8: Reliability and validity tests need to be supplemented.

Response 8: The requested information was detailed in section 2.2 within the paragraphs of each instrument.

Comment 9: In the results section, what is the purpose of analyzing the impact of coping on overall perceptions of stress and each dimension separately? This is not a usual way of reporting.

Response 9: Thank you for pointing this out. We have added the following information in the 2.3. Analysis section to clarify this observation: 'We performed these three types of analysis to be more specific. In addition to analyzing the relationship between the different coping strategies and the total perceived stress scale, we also examined their relationship with the two factors of the scale to understand how each factor is associated with the various coping strategies.' This information can be found on lines 190 to 193.

Furthermore, we would like to highlight that the analysis of the effects of coping strategies on general perceptions of stress and each specific dimension is based on several fundamental factors:

Stress is complex and has many facets, including emotional, cognitive, behavioral, and physiological aspects. These dimensions can impact people's health and well-being in various ways. Therefore, analyzing the effects of coping strategies on each dimension of stress, as well as on overall stress, provides a more comprehensive and nuanced understanding of how coping mechanisms influence the experience of stress.

The impact of different types of coping strategies varies. Emotion-focused coping may be more effective in managing anxiety, while problem-focused coping may be more effective in reducing task-related stress. When each dimension is analyzed, it is possible to determine which coping techniques work best to manage each specific aspect of stress.

The design of psychological interventions, being more specific and efficient, may depend on understanding how coping affects different dimensions of stress.

This information has been added to the discussion section, specifically on lines 359 to 370.

Comment 10: The discussion section suggests an exploration of both theoretical and practical implications.

Response 10: We appreciate your suggestion and comment. Accordingly, we have added theoretical and practical implications of our study to the discussion section. This information can be found on lines 386 to 408.

We once again thank you for your valuable suggestions and are confident that these improvements will significantly strengthen our work. We remain attentive to any additional comments you may have.

Sincerely,

Roberto Zegarra.

Round 2

Reviewer 1 Report

Comments and Suggestions for Authors

I thank the authors for addressing all comments raised in the review. My impression is that the revisions improved the quality of the manuscript. However, there are some open points that should be addressed. 

(1) Authors responded that they want to stick with interpreting the cut-offs for model fit on basis of data derived for continuous normally distributed data estimated with the maximum-likelihood method. Again, I want to stress that this introduces errors (see e.g., She & Maydeu-Olivares, 2020). Ignoring this problem biases the interpretation of the findings from the CFA.  

Shi, D., & Maydeu-Olivares, A. (2020). The effect of estimation methods on SEM fit indices. Educational and Psychological Measurement80(3), 421-445.

(2) Thank you for clarifying that you accept the model containing three correlated factors. I think a part of the unclarity stems from the point that you only report a single omega coefficient for the model. Considering that you assume three factors that translate into three manifest sub scales, an omega value must be computed and reported for each factor. 

(3) Please adjust the language when describing and discussing your findings with regard to the cross-sectional nature of the data. Often, you describe them as if they were longitudinal, which is not the case. Further, this type of data does not allow for conclusions on causality (e.g., "effects on" etc.) or within-person processes (e.g., "we found that as participants’ age increased, their

perception of stress decreased significantly"). 

Comments on the Quality of English Language

Professional proofreading is recommended

Author Response

Dear Reviewer, we sincerely appreciate your comments on our manuscript. Below, we present our responses to each of your observations:

Comment 1: Authors responded that they want to stick with interpreting the cut-offs for model fit on basis of data derived for continuous normally distributed data estimated with the maximum-likelihood method. Again, I want to stress that this introduces errors (see e.g., She & Maydeu-Olivares, 2020). Ignoring this problem biases the interpretation of the findings from the CFA.  

Shi, D., & Maydeu-Olivares, A. (2020). The effect of estimation methods on SEM fit indices. Educational and Psychological Measurement80(3), 421-445.

Response 1: We completely agree with your observation, and we sincerely appreciate the valuable reference you provided. Indeed, as highlighted in the referenced article, under the DWLS estimator, the CFI indicator tends to be overestimated, and the RMSEA indicator tends to be underestimated, with the SRMR being the most robust in this situation. In our model, the SRMR without correlating the residuals was very close to 0.08, and our CFI and RMSEA also did not indicate a good fit. Therefore, it is reasonable to believe that the model still did not have an adequate fit, considering your observations. This justifies our decision to correlate the model's residuals, allowing the SRMR to reach a stricter fit value. In response to your comment, we have revised the data analysis section to address this issue. This additional information can be found in lines 177 to 179

Comment 2: Thank you for clarifying that you accept the model containing three correlated factors. I think a part of the unclarity stems from the point that you only report a single omega coefficient for the model. Considering that you assume three factors that translate into three manifest sub scales, an omega value must be computed and reported for each factor. 

Response 2: We agree. We have estimated and reported the omega coefficient for each factor. This added information can be found in lines 236 to 238.

Comment 3: Please adjust the language when describing and discussing your findings with regard to the cross-sectional nature of the data. Often, you describe them as if they were longitudinal, which is not the case. Further, this type of data does not allow for conclusions on causality (e.g., "effects on" etc.) or within-person processes (e.g., "we found that as participants’ age increased, their perception of stress decreased significantly"). 

Response 3: Thank you for pointing this out. We have corrected the language in the manuscript to appropriately reflect the cross-sectional nature of the data. Specifically, we have revised any language that implied a longitudinal design or causality, ensuring the descriptions are consistent with the limitations of cross-sectional data.

We once again thank you for your valuable suggestions. We remain attentive to any additional comments you may have.

Sincerely,

Roberto Zegarra

Reviewer 2 Report

Comments and Suggestions for Authors

The manuscript has been greatly improved and basically meets the requirements for publication. 

Author Response

Dear Reviewer

We are grateful to you for your time in helping us to advance with this opportunity for sharing the findings of our study.

Sincerely,

Roberto Zegarra